# Novel Metabolites as Potential Indicators of Recovery After Large Vessel Occlusion Stroke: A Pilot Study

**DOI:** 10.3390/neurolint17020030

**Published:** 2025-02-18

**Authors:** Evgeny V. Sidorov, Kyle Smith, Chao Xu, Dharambir K. Sanghera

**Affiliations:** 1Department of Neurology, College of Medicine, University of Oklahoma Health Sciences Center, 920 S.L Young Blvd. #2040, Oklahoma City, OK 73104, USA; 2Oklahoma Center for Neuroscience, University of Oklahoma Health Sciences Center, Oklahoma City, OK 73104, USA; 3Department of Biostatistics and Epidemiology, University of Oklahoma Health Sciences Center, Oklahoma City, OK 73104, USA; 4Department of Pediatrics, College of Medicine, University of Oklahoma Health Sciences Center, 940 Stanton Y Blvd, BMSB 317D, Oklahoma City, OK 73104, USA; 5Harold Hamm Diabetes Center, University of Oklahoma Health Sciences Center, Oklahoma City, OK 73104, USA; 6Department of Physiology, College of Medicine, University of Oklahoma Health Sciences Center, Oklahoma City, OK 73104, USA; 7Department of Pharmaceutical Sciences, College of Pharmacy, University of Oklahoma Health Sciences Center, Oklahoma City, OK 73104, USA

**Keywords:** stroke outcomes, outcomes, biomarkers, metabolome, NMR, LC-MS

## Abstract

**Introduction:** Serum metabolome changes after acute ischemic stroke (AIS), but the significance of this is poorly understood. We evaluated whether this change is associated with AIS outcomes in patients with large vessel occlusion (LVO). To improve validity, we combined cross-sectional and longitudinal designs and analyzed serum using Nuclear Magnetic Resonance (NMR) and Liquid Chromatography–Mass Spectrometry (LC-MS). **Methodology:** In the cross-sectional part, we compared serum metabolome from 48 LVO strokes, collected at 48–72 h, and analyzed with NMR, while in the longitudinal part, we compared metabolome from 15 LVO strokes, collected at <24 h, 48–72 h, 5–7 days, and 80–120 days, and analyzed with LC-MS between patients with modified Rankin Scores (mRS) of 0–3 and 4–6 at 90 days. We hypothesized that compounds elevated in patients with mRS 0–3 in the cross-sectional part would also be elevated in the longitudinal part, and vice versa. We used regression for the analysis and TSBH for multiple testing. **Results:** In the cross-sectional part, cholesterol, choline, phosphoglycerides, sphingomyelins, and phosphatidylethanolamines had lower levels in patients with an mRS of 0–3 compared to an mRS of 4–6. In the longitudinal part, lower levels of sphingomyelin (d18:1/19:0, d19:1/18:0)* significantly correlated with an mRS of 0–3 in patients with small infarction volume, while lower levels of sphingolipid N-palmitoyl-sphingosine (d18:1/16:0), 1-palmitoyl-2-docosahexaenoyl-GPC (16:0/22:6), 1-palmitoyl-2-docosahexaenoyl-GPE, palmitoyl-docosahexaenoyl-glycerol (16:0/22:6), campesterol, and 3beta-hydroxy-5-cholestenoate correlated with an mRS of 0–3 in patients with large infarction volume. **Conclusions:** This pilot study showed that lower levels of lipidomic components nerve cell membrane correlate with good AIS outcomes. If proven on large-scale studies, these compounds may become important AIS outcome markers.

## 1. Introduction

Acute ischemic stroke (AIS) is a leading cause of serious disability in the United States [1]. The annual cost of stroke care totals USD 34 billion and continues to grow [2]. Novel diagnostic and treatment techniques significantly improved AIS outcomes, but their prediction remains uncertain and purely based on clinical and radiologic data. Serum biomarkers are well developed and actively used in many areas of medicine, but nonexistent in AIS. The isolation of the central nervous system (CNS) from other body compartments by the blood–brain barrier (BBB) prevents serum from changing in many CNS disorders. The interruption of the BBB after AIS leads to free communication between the CNS and the serum compartments [3]. This may be used for the identification of AIS serum markers. Metabolomics is a novel method that investigates disease pathophysiology and carries the ability to identify many low-molecular-weight metabolites in bodily fluids. Such metabolites may serve as disease biomarkers. Many investigators found an important association of serum metabolites with AIS [4,5]. They associated these changes with post-stroke depression [6], cognitive impairment [7], and overall disease severity [8,9]. These findings, however, were often contradictory, and explanations were speculative. Our group identified common trends in the change in serum metabolome after AIS and focused on finding clinical applications for these changes. In this pilot study, we evaluated whether trends of serum metabolome are associated with outcomes of Large Vessel Occlusion (LVO) stroke. We used different technologies for metabolome analysis: Nuclear Magnetic Resonance (NMR) and Liquid Chromatography–Mass Spectroscopy (LC-MS) and a combination of cross-sectional and longitudinal study designs to improve confidence in our findings.

## 2. Methods

### 2.1. Study Goal and Design

The goal of this study was to evaluate whether changes in serum metabolome are associated with AIS outcomes. We evaluated serum metabolome on 2 sets of patients using cross-sectional and longitudinal analyses. In the cross-sectional part of this study, we evaluated the difference in serum metabolome at 48–72 h between patients with an mRS of 0–3 (good outcomes) and 4–6 (poor outcomes) at 3 months; in the longitudinal part, we evaluated the difference in the trajectory of serum metabolome change at 4 time-points: (1) <24 h; (2) 48–72 h; (3) 5–7 days; and (4) during the follow-up clinic visit (80–120 days) between patients with mRS 0–3 and 4–6 at 3 months. All collected samples were kept at −80 degrees Celsius, then processed and analyzed at the same time. Our hypothesis was that serum metabolome compounds associated with AIS outcomes would have similar changes in the cross-sectional and longitudinal parts of this study: (1) metabolites that elevated in patients with good/poor outcomes in the cross-sectional part would also elevate in the longitudinal part; (2) metabolites that decreased in patients with good/poor outcomes in the cross-sectional part would also decrease in the longitudinal part. Given that our longitudinal cohort was heterogeneous and included patients with and without successful reperfusion, we also performed analysis for patients with small (<40 mL) and large (>40 mL) infarction volume.

The study participants were a part of our ongoing, institutional board-approved, Metabolome in Ischemic Stroke Study (MISS), which investigates biomarkers using genomics, metabolomics, and other -omics technologies [4,10,11]. All participants signed their informed consent. We included patients with internal carotid, anterior, middle, or posterior cerebral artery occlusion; magnetic resonance imaging (MRI) was performed for AIS definition. Two independent investigators calculated AIS volumes using MR segmentation software [12]. This software utilizes an interactive computer-aided detection scheme, where the user identifies and draws the stroke boundary for each of the identified stroke slices, and the computer-aided detection scheme computes the infarction volume. At presentation, we evaluated patients with the National Institute of Health Stroke Scale (NIHSS) and pre-admission modified Rankin Scale (mRS). During the follow-up visit (80–120 days), we performed neurological exams, reviewed medical records for recurrent strokes, and again carried out mRS evaluations. We excluded patients with (1) hemorrhagic conversion of AIS and development of parenchymal hematoma [13]; (2) systemic infection; (3) renal disease with glomerular filtration rate <45; and (4) recurrent stroke during follow-up visits because these conditions could show distinct metabolomic profiles and confound our results.

### 2.2. Metabolome Analysis

We have chosen to use different analytic platforms in the cross-sectional and the longitudinal parts of this study to improve the validity of our findings. In the cross-sectional part, serum metabolome was analyzed using NMR, which provides information on 250 fundamental serum metabolites, well established and commonly used for AIS metabolomic research, including large data repositories such as UK Biobank (UKBB). In the longitudinal part, serum metabolome was analyzed using LC-MS, which provides more detailed information on 1554 metabolites (for example, while NMR shows the level of sphingomyelin, in LC-MS sphingomyelin, it is represented by 29 sphingomyelin derivatives) but is more expensive and less commonly used and has fewer AIS publications. Observing similar findings across NMR and LC-MS platforms would improve the validity of our results.

NMR metabolomics platform (Nightingale Health Ltd., Helsinki, Finland) was used to quantify the circulating metabolites of non-lipidomic and lipidomic origins. This high-throughput metabolomics platform provides the simultaneous quantification of amino acids, ketone bodies, glycol high-density metabolites, routine lipids, abundant fatty acids, and different lipoprotein subclasses, including high-density lipoproteins (HDLs); intermediate-density lipoproteins (IDLs); low-density lipoproteins (LDLs); and very-low-density lipoproteins (VLDLs)/chylomicrons in absolute concentration units. The NMR platform has been applied extensively in epidemiological studies [14,15], and details of these experiments are described elsewhere [16,17].

For the LC-MS metabolomics platform (Metabolon, Inc. Morrisville, NC, USA), the serum was separated from the red cap vacutainer by centrifugation at 2200 rpms for 15 min, after allowing the blood to clot for at least 30 min in a vertical position before spinning. EDTA plasma samples were extracted, proteins were precipitated with methanol, and aliquots were analyzed by ultra-high-performance LC-MS in the positive (one optimized for hydrophilic, the other hydrophobic compounds), negative, and polar ion modes. All methods utilized Waters ACQUITY ultra-performance liquid chromatography and Thermo Scientific Q-Exactive high resolution/accurate mass spectrometer interfaced with a heated electrospray ionization source and Orbitrap mass analyzer operated at 35,000 mass resolution. The assignment of a unique mass, signal-to-noise calculation, and compound identification were based on the mass spectral pattern compared to NIST and Wiley mass spectral libraries, followed by visual inspection for quality control [18]. For quality control and assurance, an internal matrix was used as well as several internal standards to determine instrument variability, with representative relative standard deviations of 3% (plasma) for internal standards and 8% (plasma) for endogenous biochemicals.

### 2.3. Statistical Analysis

The clinical and demographic variables were summarized using mean with standard deviation or error for continuous variables and numbers and percentages for categorical variables. Raw metabolome profile data were normalized through log-transformation. The analysis was performed in R4.4.2. We used multivariate linear regression for the cross-sectional part and linear mixed regression model for the longitudinal part of this study. We used age, gender, and race as covariates for cross-sectional and longitudinal analysis. We used the Two-Stage Benjamini–Hochberg (TSBH) procedure for multiple testing in order to minimize false positive correlations and, again, increase confidence in our results. We set alpha significance at <0.05.

## 3. Results

The cross-sectional part included 48 LVO stroke patients, 34 had an mRS of 0–3 and 14 had an mRS of 4–6. The longitudinal part included 15 LVO stroke patients. Demographic and clinical characteristics for both parts are presented in Table 1.

In the cross-sectional part, out of 250 compounds, 114 were significantly different in patients with an mRS of 0–3 compared to an mRS of 4–6 (Appendix A). All 114 metabolites had lower levels associated with good outcomes and represented a metabolism of different lipoprotein sub-classes (100 compounds), saturated and unsaturated fatty acids (6 compounds), energy metabolites (glucose and lactate), cholesterol, choline, phosphoglycerides, sphingomyelins, and phosphatidylethanolamines.

In the longitudinal part, the analysis of all 15 patients did not show metabolites associated with clinical outcomes. We identified several significant metabolites when analyzed patients with small and large infarction volume. In seven patients with small infarction volume (average infarction volume 7.99 ± 6.79 mL, follow-up mRS was 1.57 ± 2.23), we observed that lower levels of sphingomyelin (d18:1/19:0, d19:1/18:0) * correlated with good outcomes, while higher levels correlated with poor outcomes, particular at <24 h time point (*p* = 0.02) (Table 2). In eight patients with large infarction volume (average infarction volume 97.31 ± 49.06 mL, follow-up mRS was 3.5 ± 2.33), we observed that lower levels of six metabolites significantly correlated with good outcomes, while higher levels correlated with poor outcomes: one derivative of sphingolipids N-palmitoyl-sphingosine (d18:1/16:0) (*p* = 0.03); two derivatives of phosphatidylethanolamine: 1-palmitoyl-2-docosahexaenoyl-GPC (16:0/22:6) (*p* = 0.03) and 1-palmitoyl-2-docosahexaenoyl-GPE (*p* = 0.02); one derivative of glycerol: palmitoyl-docosahexaenoyl-glycerol (16:0/22:6) (*p* = 0.03); and two derivatives of steroids: campesterol (*p* < 0.01) and 3beta-hydroxy-5-cholestenoate (*p* = 0.03) (Table 2).

## 4. Discussion

In this study, we identified seven candidate markers of AIS outcomes, five of these markers showed corroborated changes in cross-sectional and longitudinal analysis. All these compounds belong to phospholipids and represent important components of the neural cell membrane. It is important that good AIS outcomes were associated with lower levels of all compounds in cross-sectional and longitudinal analysis, which may indicate the degree of neuronal membrane damage. If such hypothesis is true, there is a possibility that infarcted areas seen on diffusion-weighted MRI still contain functioning nerve cells, which may facilitate recovery.

Although the exact mechanism of change in serum metabolome after AIS is unknown, conventional explanations include diffusion through the interrupted BBB [3] and the activation of the ischemia-mediated pathways [19,20]. These explanations correspond to the findings in our study, particularly for the patients with large infarction volumes. These patients have extensive areas of BBB interruption, which led to a well-organized difference in the levels of six metabolites, observed through multiple time points. In patients with small infarction volume, the mechanism of change in metabolome may be different. These patients have small areas with BBB interruption, reflecting a change in only one metabolite at <24 h (Table 2). In the combined analysis of small- and large-volume patients, these findings could have canceled each other out. A separate analysis for large and small volumes was not possible in the cross-sectional group because all patients had similar infarction volumes.

The change in serum metabolome after stroke may be also related to the ischemia-mediated inflammation [4,21,22]. This reaction may or may not be associated with the elevation in traditional inflammatory markers such as C-reactive protein and erythrocyte sedimentation rate. We did not observe a correlation of inflammatory markers with the outcomes in our study, which speaks against such a mechanism. On the other hand, we observed a correlation of good outcomes with lower levels of steroid derivatives in the longitudinal part of this study. This suggests that the diminished inflammatory response may be associated with better recovery. However, the lack of such data in the cross-sectional analysis limits our conclusions. The decrease in lipidomic compounds after AIS may also be related to the poor nutritional status of patients. This happens because AIS patients commonly develop dysphagia, and feeding through nasogastric tubes is not initiated timely. Although this could have affected the study results, poor nutrition status cannot explain the better recovery observed after AIS.

Searching through public metabolome databases, including Cayman, Avanti, PubChem, and others, could not identify the pathophysiological background for the observed changes. N-palmitoyl-sphingosine (d18:1/16:0), also known as Ceramide C16, played role in forming channels in mitochondrial membrane, increased levels in apoptosis, and acute coronary syndrome [23]; 1-palmitoyl-2-docosahexaenoyl-GPC (16:0/22:6) was associated with atherosclerotic plaques [24]. Steroid derivative 3beta-hydroxy-5-cholestenoate was related to the pathophysiology of Alzheimer disease [25]; while Campesterol serves as a steroid precursor [26].

If proven on a larger cohort, the results of our study may be used for the prognostication of ischemic stroke outcomes. Despite the general understanding that the diffusion-weighted sequence of MRI represents permanent brain damage, some case reports indicate the reversibility of such an injury and the presence of viable brain cells within the infarcted area [27,28]. In this case, the association of lower metabolite levels with better functional recovery could indicate the number of viable cells within infarcted area and explain reasonable recovery in some patients with comparatively large strokes. A gradual change in metabolites after ischemic stroke suggests the continuous transformation of ischemic penumbra into the core or metabolic changes within the core even days after the index event. A better understanding of this pathophysiology will not only allow better prediction of outcomes but may also lead to the development of neuroprotective agents in the future [29].

The strengths of our study include the homogeneous patient population of LVO strokes, and the unique study design where serum metabolome was evaluated using NMR and LC-MS technology in separate groups of patients using cross-sectional and longitudinal study designs. This marks out our study among many others that simply compared metabolome between stroke patients and controls. The most important finding of our study is that lower levels of lipidomic compounds were uniformly associated with better outcomes across different analytic platforms and study designs. Limitations include a comparatively small sample size and wide variation in the infarction volumes in the longitudinal cohort. Also, the difference between NMR and LC-MS panels did not allow for evaluating all compounds. It is important to stress that the results of this study should be viewed as corroborated changes in the cross-sectional and longitudinal parts along with significant findings after TSBH application. Separate findings from either part of this study are not convincing. Our findings need to be validated on large-scale studies before any final conclusions are made.

## Figures and Tables

**Table 1 neurolint-17-00030-t001:** Baseline patients’ characteristics. ^1^ Mean (SD); n (%).

Characteristic	Cross-Sectional Part; n = 48	Longitudinal Part; n = 15 ^1^
Age	62.5 (15.5)	64.73 (14.18)
(Sex) Female	23 (48%)	7 (46.7%)
(Race) White	34 (71%)	10 (66.7%)
Black	11 (23%)	3 (20.0%)
Other	3 (6%)	2 (13.4%)
NIHSS at presentation	8.85 (7.0)	14.27 (6.71)
Pre-adm mRS	0.81 (1.16)	0.33 (0.82)
mRS 3 months	2.81 (1.65)	2.60 (2.41)
Infarction volume	24.82 (6.77)	55.63 (57.88)
Hx of HTN	34 (71%)	10 (66.7%)
Hx of HLD	22 (45.8)	7 (46.7%)
Hx of CAD	10 (21%)	3 (20.0%)

**Table 2 neurolint-17-00030-t002:** Change in metabolites after AIS for good and poor outcomes.

Cross-Sectional Part Metabolite (NMR) Changes 48–72 h	Corresponding Longitudinal Part Metabolite Derivative Trend (LC-MS)	Trends for Patients with Good (mRS of 0–3) and Poor (mRS of 4–6) Outcomes
Sphingomyelin (small infarction volume) Decreased in patients with good compared to poor outcomes (R = −17.59 ± 7.28; *p* = 0.03)	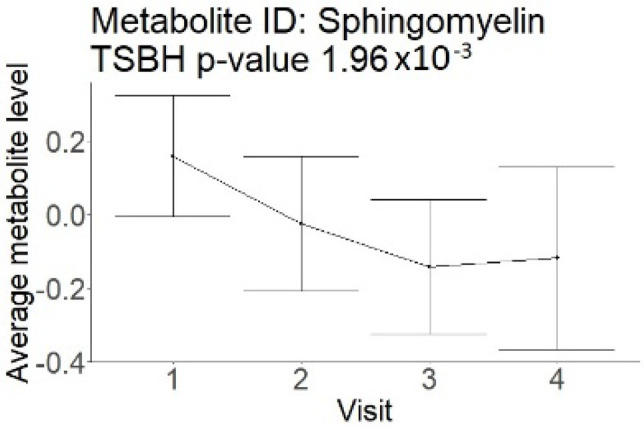	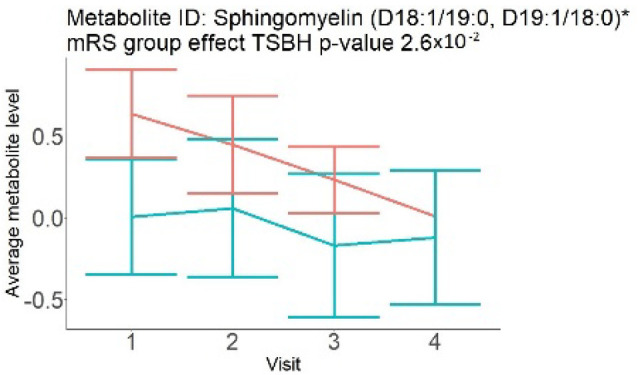
Ceramides (not a part of NMR panel)	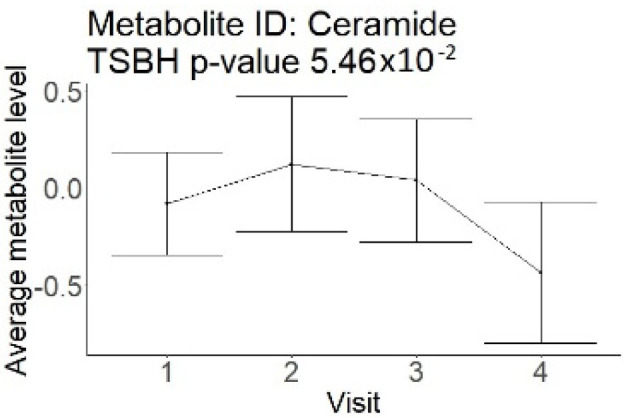	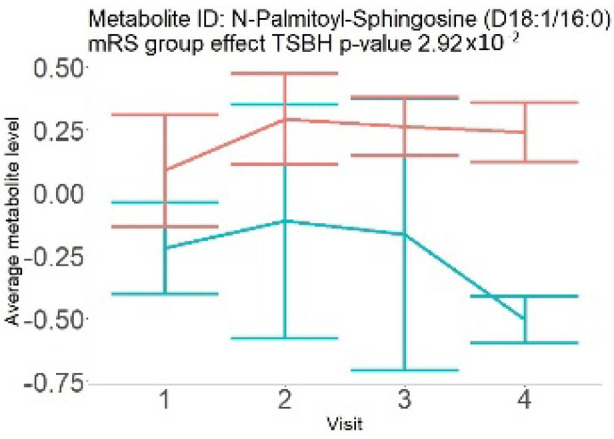
Phosphatidylethanolamine (large infarction volume) Decreased in patients with good compared to poor outcomes (R = − 4.30 ± 1.70; *p* = 0.02) Phosphoglycerides (large infarction volume) Decreased in patients with good compared to poor outcomes (R = − 4.13 ± 1.61; *p* = 0.02)	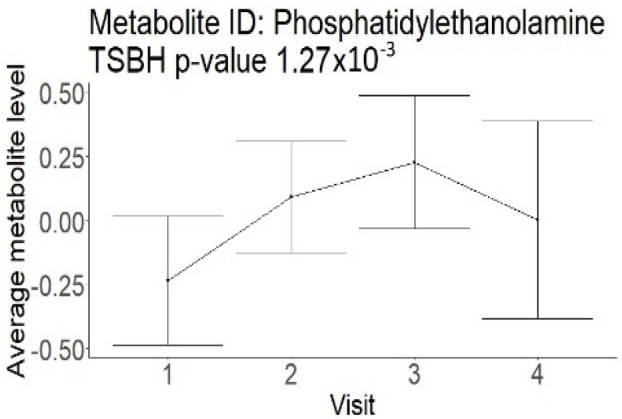	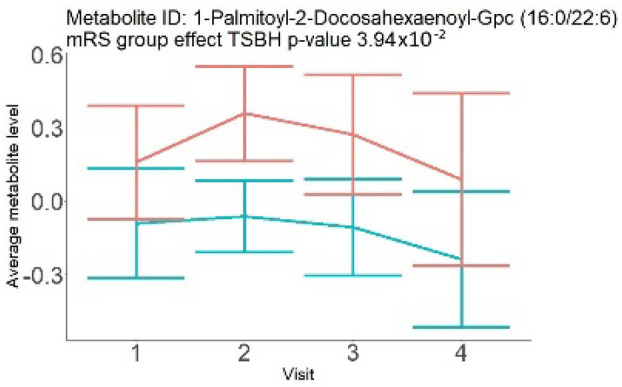
		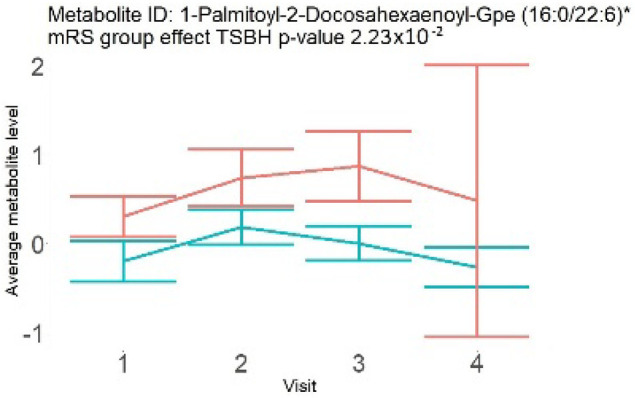
Glycerol (large infarction volume) Insignificantly decreased in patients with good compared to poor outcomes (R = − 6.72 ± 4.20; *p* = 0.12)	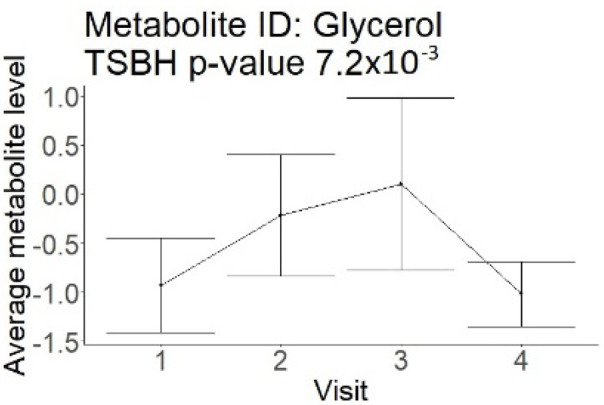	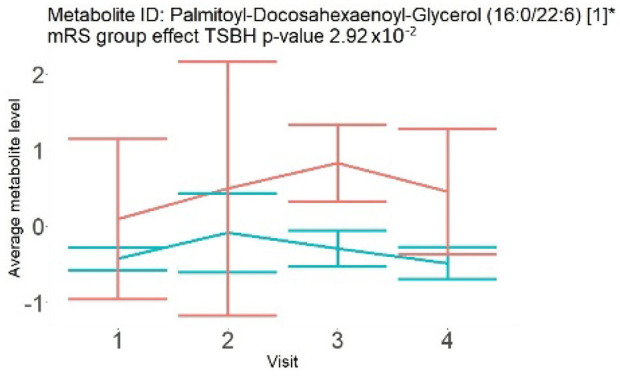
Steroids (not a part of NMR panel)	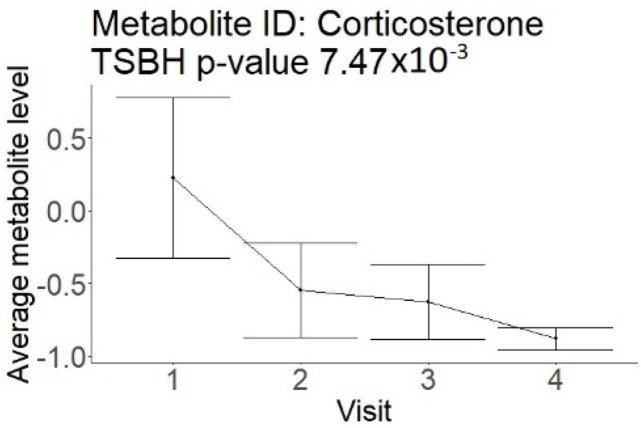	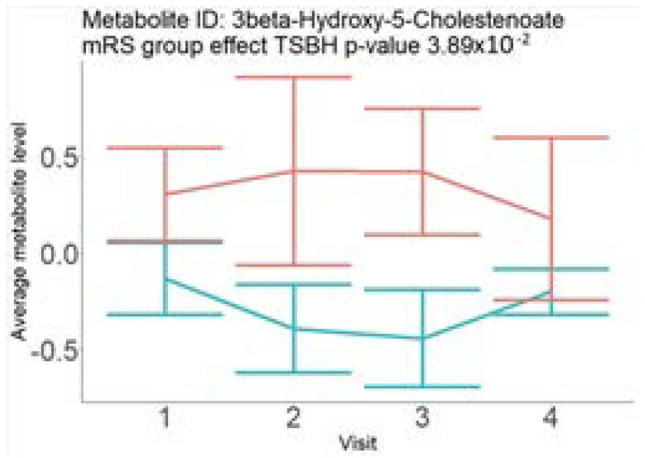
		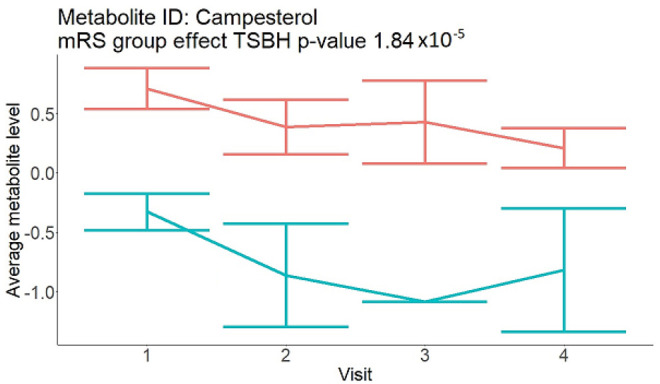

Visti time points: (1) <24 h; (2) 48–72 h; (3) 5–7 days; and (4) during follow-up clinic visit (80–120 days); Green line represents metabolite levels with good (0-3), Red line represents metabolite levels with poor (4-6) mRS at 3 months.

## Data Availability

All data are available upon request.

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
