# Peer review of "Novel Metabolites as Potential Indicators of Recovery After Large Vessel Occlusion Stroke: A Pilot Study"

_2035-8377, 2025, doi:10.3390/neurolint17020030_

Round 1

Reviewer 1 Report

Comments and Suggestions for Authors

The sample size for this study is very small, how did the authors decide on it?

Surprisingly even though the infarct size was larger in the longitudinal cohort, the MRS was better at 3/12

Did the authors divide into posterior and anterior circulation?

I notice that there is no mortality and symptomatic ICH as outcomes? These are common in LVO strokes.

At which time point was the serum drawn, was this kept uniform throughout the study? One of the major problems with metabolomic studies is that the results vary heavily with collection at different time points and the processing of the sample.

Author Response

The sample size for this study is very small, how did the authors decide on it?

Thank you for your comment. This study has a particularly small N on the longitudinal part, however, it is positioned as pilot (pointed in the heading) and does not make any final conclusions. In this study we bring an idea that lower levels of lipidomic metabolome may be associated with better outcomes after LVO stroke. In addition, study has 2 components: cross-sectional and longitudinal. The cross-sectional part includes 48 participants which is a reasonable size for pilot study. The longitudinal part changes corroborate with cross-sectional part (have the same direction of change), suggesting that our hypothesis is true. We added in the Discussion section that the results should only be considered as combination cross-sectional and longitudinal parts of the study and must be reproduced. In addition, we used TSBH procedure for multiple testing which showed significant p-values. This also puts confidence in our results. All this was added to the Discussion section.

Surprisingly even though the infarct size was larger in the longitudinal cohort, the MRS was better at 3/12

Thank you for your comment. This observation may be related to the patient population. Patients recruited in the cross-sectional cohort have more homogeneous infarction volumes, this group also has more narrow standard deviation. In the longitudinal cohort patients would either have very large or small infarction volume depending on the success of endovascular thrombectomy. We put this as a limitation of our study. 

Did the authors divide into posterior and anterior circulation?

We included patients with internal carotid, anterior, middle or posterior cerebral artery occlusion as it says in the Methods section of the manuscript.

 I notice that there is no mortality and symptomatic ICH as outcomes? These are common in LVO strokes.

Thank you for your comment. We excluded patients with symptomatic hemorrhages as stated in Methods. Symptomatic ICHs such as PH 2 by Heidelberg classification could confound our metabolome findings.

At which time point was the serum drawn, was this kept uniform throughout the study? One of the major problems with metabolomic studies is that the results vary heavily with collection at different time points and the processing of the sample.

This is a very insightful comment. I agree that metabolome is very dependent on many factors and time of the sample collection is one of them. In our study all samples were collected uniformly as mentioned in Methods and frozen at -80 C. In the cross-sectional part samples were collected 48-72 hours, and in the longitudinal part they were collected <24 hours, 48-72 hours, 5-7 days, 80-120 days. All samples were analyzed at the same time. We modified the description of this process in Methods for clarity.

Reviewer 2 Report

Comments and Suggestions for Authors

The manuscript investigates the role of serum metabolites as potential biomarkers for recovery in patients with large vessel occlusion (LVO) stroke. By combining cross-sectional and longitudinal designs, the authors identify lipidomic compounds that may correlate with functional outcomes, as measured by the modified Rankin Scale (mRS). The study utilizes Nuclear Magnetic Resonance (NMR) and Liquid Chromatography-Mass Spectrometry (LC-MS) platforms, showcasing their complementary strengths.

1.        While candidate biomarkers are identified, the mechanistic links between these metabolites and stroke recovery are underexplored.

2.        The variability in infarction volumes and reperfusion success complicates the interpretation of metabolomic changes.

3.        The manuscript does not sufficiently discuss how the findings might translate into clinical practice or guide therapeutic interventions.

4.        The choice of statistical methods (e.g., TSBH procedure) is appropriate but would benefit from a clearer explanation for readers unfamiliar with metabolomics.

5.        The discussion could better integrate findings from similar studies to highlight the novelty and importance of the results.

6.        Minor grammatical errors and inconsistencies in phrasing, such as “yield distinct metabolomic profiles,” require editing for clarity.

Author Response

The manuscript investigates the role of serum metabolites as potential biomarkers for recovery in patients with large vessel occlusion (LVO) stroke. By combining cross-sectional and longitudinal designs, the authors identify lipidomic compounds that may correlate with functional outcomes, as measured by the modified Rankin Scale (mRS). The study utilizes Nuclear Magnetic Resonance (NMR) and Liquid Chromatography-Mass Spectrometry (LC-MS) platforms, showcasing their complementary strengths.

  1. While candidate biomarkers are identified, the mechanistic links between these metabolites and stroke recovery are underexplored.

Thank you for your comment. We elaborated more on other possible connections between lower levels of the metabolites and better functional outcomes, including more details. Besides preservation of the nerve cell membrane, it may be related to stroke mediated inflammatory reaction or even nutritional deficiencies secondary to dysphagia. It is now included in the Discussion.

  1. The variability in infarction volumes and reperfusion success complicates the interpretation of metabolomic changes.

I agree with this very insightful comment. This is particularly true for the longitudinal cohort where the infarction volumes were either very large or very small. They also show wider standard deviations. We put this as a limitation of our study.

  1. The manuscript does not sufficiently discuss how the findings might translate into clinical practice or guide therapeutic interventions.

Thank you for pointing this out. If results of this study are validated in a larger sample they may be used for prognostication of the outcomes or even for development of neuroprotective medications which can stabilize nerve cell membranes. We reflected it in the Discussion.

  1. The choice of statistical methods (e.g., TSBH procedure) is appropriate but would benefit from a clearer explanation for readers unfamiliar with metabolomics.

It clarified it in Methods.

  1. The discussion could better integrate findings from similar studies to highlight the novelty and importance of the results.

Thank you for your comment. I think this is related to your comment number #3. We added additional references to the Discussion and elaborated on the importance of these findings.   

  1. Minor grammatical errors and inconsistencies in phrasing, such as “yield distinct metabolomic profiles,” require editing for clarity.

This has been corrected, the manuscript was proofread.

Round 2

Reviewer 2 Report

Comments and Suggestions for Authors

The authors have addressed my all queries and improved the overall quality of their manuscript which is now suitable for acceptance.